# Pushing the Limits of Thermal Resistance in Nanocomposites: A Comparative Study of Carbon Black and Nanotube Modifications

**DOI:** 10.3390/nano15070546

**Published:** 2025-04-03

**Authors:** Johannes Bibinger, Sebastian Eibl, Hans-Joachim Gudladt, Bernhard Schartel, Philipp Höfer

**Affiliations:** 1Institute of Lightweight Engineering, University of the Bundeswehr Munich, 85577 Neubiberg, Germany; philipp.hoefer@unibw.de; 2Bundeswehr Research Institute for Materials, Fuels and Lubricants (WIWeB), 85435 Erding, Germany; sebastianeibl@bundeswehr.org; 3Institute of Materials Science, University of the Bundeswehr Munich, 85577 Neubiberg, Germany; hans-joachim.gudladt@unibw.de; 4Bundesanstalt für Materialforschung und -Prüfung (BAM), Unter den Eichen 87, 12205 Berlin, Germany; bernhard.schartel@bam.de

**Keywords:** nanomaterial, cabon fiber-reinforced polymer (CFRP), carbon black (CB), carbon nanotube (CNT), improving heat conduction, prediction thermal properties

## Abstract

Enhancing the thermal resistance of carbon fiber-reinforced polymers (CFRPs) with flame retardants or coatings often leads to increased weight and reduced mechanical integrity. To address these challenges, this study introduces an innovative approach for developing nanocomposites using carbon-based nanoparticles, while preserving the structural lightweight properties. For this, carbon black particles (CBPs) up to 10% and carbon nanotubes (CNTs) up to 1.5% were incorporated into the RTM6/G939 composite material. The obtained samples were then analyzed for their properties and heat resistance under one-sided thermal loading at a heat flux of 50 kW/m^2^. Results demonstrate that integrating these particles improves heat conduction without compromising the material’s inherent advantages. As a result, thermo-induced damage and the resulting loss of mechanical strength are delayed by 17% with CBPs and 7% with CNTs compared to the unmodified material. Thereby, the thermal behavior can be accurately modeled by a straightforward approach, using calibrated, effective measurements of the nanoparticles in the polymer matrix rather than relying on theoretical assumptions. This approach thus provides a promising methode to characterize and improve thermal resistance without significant trade-offs.

## 1. Introduction

Carbon fiber-reinforced polymers (CFRP) are composite materials that are characterized by their high strength and low weight [1,2]. Compared to lightweight materials such as aluminium alloys, they even provide up to seven times the tensile strength at almost half the weight [1,3,4,5,6]. These outstanding properties make CFRP particularly attractive for use in aircraft construction [7,8,9,10,11], especially as central components such as wings, fuselages, tail units and various panelling elements [3,10]. In their operating environment, these parts are exposed to various thermal loading scenarios. These range from heating by solar radiation [12] or turbines [13], the overheating of electrical equipment, cables or batteries [14,15] to fires [12,16,17,18] or lightning strikes [19]. In military contexts, there are additional threats from flame weapons such as molotov cocktails [12,20,21,22], projectile hits [12], heat flashes [23,24,25] or laser weapons [26,27]. In contrast to the alternative metallic structures, however, these composite materials have a lower thermal resistance due to the polymer matrix [5,28,29,30]. Under specific loading conditions [31], the operating temperature of just 120 °C [32] for the RTM6 matrix, for example, can quickly be exceeded. As a result, thermal damage can occur, which significantly impairs the mechanical properties of the material [33,34,35].

To protect CFRP from thermal exposure, various approaches have therefore been developed. One option is to add flame retardants to the material [36,37,38]. Inorganic flame retardants, such as metal hydroxides or boron compounds, release water during their decomposition, which cools the material and dilutes the pyrolysis gases at the same time [39]. Additionally, a glassy crust forms on the surface, acting as a protective layer [39]. Unfortunately, the use of larger quantities of these additives can increase the material weight and reduce the mechanical strength of the components [40]. Another approach is the integration of fire-protective interlayers made of, for example, ceramic or polyetherimide into CFRP [41,42]. These thermally insulating layers act as sacrificial barriers under thermal stress and reduce the heat conduction to the CFRP structure while delaying the propagation of the fire deep into the material. This significantly delays the time to mechanical failure, although an increased mass and a larger cross-section must be accepted. Furthermore, special coatings offer protection against thermal influences. Intumescent coatings, for example, expand under heat and thus form a thermally insulating barrier [36,38,43]. In contrast, polyurethane-based top coats with white silicate fillers can increasingly reflect the impinging heat radiation and thus delaying the thermal decomposition of the material [31,44,45]. However, these coatings can also be spontaneously pyrolyzed and ablated under certain conditions, while simultaneously increasing the weight of the component without contributing to its load-bearing capacity.

Thus, the benefits of these traditional methods are often accompanied by increased weight and reduced mechanical strength. In aerospace applications, where maintaining low mass and structural integrity is crucial, the challenge lies in improving the thermal performance of the material without sacrificing these key properties. Recently, polymer-based nanocomposites of carbon black particles (CBPs) [46] and carbon nanotubes (CNTs) [35] have gained attention as promising solutions. Their high thermal conductivity, low weight, and ability to form heat-dissipating networks provide an effective way to enhance CFRP’s thermal performance without the typical drawbacks of conventional approaches. However, comparative studies on these carbon-based modifications are still lacking that quantify the effect of these nanoparticles on heat conduction, thermo-induced damage, and mechanical properties under thermal stress. In this study, nanocomposites based on carbon fiber-reinforced polymers with CBPs and CNTs were therefore fabricated. By systematically increasing the particle concentration, the thermal resistance was pushed to the practical limits within the material’s inherent constraints. Subsequently, the modified materials were comprehensively examined regarding their properties and thermal stability. The primary objectives are twofold: first, to develop a foundational model that deepens the understanding of heat conduction in carbon-based nanocomposites; and second, to enhance the thermal properties of these materials to prevent localized overheating and reduce internal stresses under thermal exposure. As a result, the mechanical integrity of the materials is expected to be preserved over an extended time period.

## 2. Materials

The commercially available polymer HexFlow^®^ RTM6 [32] from Hexcel Composites GmbH (Stade, Germany) is a one-component resin composed of the epoxy component Tetraglycidyl-4,4′-diaminodiphenylmethane (TGDDM) and the hardeners 4,4′-Methylenebis (2,6-diethylaniline) (MBDA) and 4,4′-Methylenebis(2-isopropyl-6-methylaniline) (MBIMA) [47]. For modification, two carbon-based nanomaterials were employed: First, PRINTEX^®^ L6 carbon black particles (CBPs) from Orion Engineered Carbons (Eschborn, Germany), characterized by a specific surface area of 21–1200 m^2^/g and no surface treatment [48]. Second, multi-walled carbon nanotubes (CNTs) of the type Baytubes^®^ C70P from Bayer MaterialScience AG (now Covestro AG, Leverkusen, Germany), with a carbon purity exceeding 95% [49]. The resin was blended with CBPs at mass concentrations of $\omega_{p,m}$ = 1, 3, 5, and 10%, or with CNTs at nominal concentrations of $\omega_{p,m}$ = 0.1, 0.5, and 1.5%, as shown in Figure 1. The particle agglomerates in these mixtures were fractured using the EXAKT 80 E three-roll mill from EXAKT Advanced Technologies GmbH (Norderstedt, Germany). The process involved two cycles with a roller gap of 5 μm, varying roller velocities, and no roller heating, over a duration of approximately 20 min [35]. The resulting suspensions were then degassed using the DAC 600.2 VAC-P speed mixer (Hauschild SpeedMixer Inc., Hamm, Germany) at a maximum frequency of 2350 rpm and a vacuum of 10 mbar at room temperature for three minutes. Subsequently, the degassed matrix was infiltrated into eight G939 fabric plies from Hexcel Composites GmbH with a fiber orientation of [(+45°/−45°)/(90°/0°)/(−45°/45°)/(0°/90°)]_s_ on a heated table at 60°C.

The resulting products were covered with Release Ease 234 TFP peel ply on their surfaces, sealed within aluminum molds, and placed in a forced convection chamber. In the first step, the system was conditioned at 160°C under a constant overpressure of 2 bar for one hour, while in the second step, the polymer matrix was completely cured at 180°C for two hours. The cured composite materials had a cross-sectional thickness of 2.3 ± 0.2 mm and a matrix mass content of approximately $\omega_{m}$ = 45%. The mass concentrations of CBPs in the laminate were $\omega_{p,1}$ = $\omega_{p,m}\cdot \omega_{m}$ = 0.5, 1.4, 2.3, and 4.5%, while the concentrations of CNTs were $\omega_{p,1}$ = 0.05, 0.2, and 0.7%. To ensure comparability among these laminates, only specimens with no detected quality defects such as pores, macroscopic delaminations or defects in fiber orientation were selected for testing using the Hill-Scan 3060 UHF ultrasonic inspection system from Dr. Hillger (Braunschweig, Germany).

## 3. Methods

### 3.1. Material Properties

To investigate the manufacturing limits of the maximum carbon-based particle concentration in the composite material, the dynamic viscosity of the polymer melt was measured using a plate-plate setup in the Haake RheoStress 600 rheometer from Thermo Electron (Karlsruhe, Germany). These tests were conducted according to DIN 53019-1 [50] at an angular velocity of 10 rad/s and a gap distance of 0.5 mm over a period of 600 s at different temperatures.

The morphology of the nanoparticles was examined using the Ultra Plus field emission scanning electron microscope (SEM) from Zeiss (Oberkochen, Germany) to capture magnified structures. An acceleration voltage of only 1 kV was applied to avoid charging effects. This enabled direct examination of particle sizes with a suitable carrier, while the samples had to be prepared to visualize the particle distribution along their cross-section [51,52].

To characterize the development of the nanoparticle network in the modified materials, electrical conductivity was analyzed in accordance with DIN EN 61340-2-3 [53]. The electrical resistance was measured using a sponge rubber electrode, a Fluke multimeter with a measuring range up to the megohm region, and a Sefelec megohmmeter up to the teraohm region, over a duration of 60 s under constant temperature and humidity conditions.

The heat transport behavior within the material was investigated with the Light Flash Analysis LFA 467 HAT HyperFlash^®^ from Netzsch (Selb, Germany) in accordance with DIN EN ISO 22007-4 [54] by measuring the thermal diffusivity, thermal conductivity and specific heat capacity. Sample density was determined using the CM224S balance from Sartorius (Göttingen, Germany) in accordance with the Archimedean principle as specified in DIN EN ISO 1183-1 [55].

The glass transition temperature of the specimens was assessed with a three-point bending setup using the Gabo Eplexor^®^ 500 dynamic mechanical analyzer from Netzsch. The samples were heated at a rate of 1 °C/min while simultaneously subjected to both static and dynamic loads at a frequency of 1 Hz.

Mechanical properties were evaluated using universal testing machines from Zwick/Roell (Ulm, Germany). The interlaminar shear strength was determined by testing 20 × 10 × 2 mm^3^ specimens according to DIN EN 2563 [56] at a traverse speed of 1 mm/min, with supports having a radius of 3 mm and a distance of 10 mm. The compressive strength of the 110 × 10 × 2 mm^3^ samples was investigated using a Celanese fixture at a velocity of 1 mm/min in accordance with DIN EN ISO 14126 [57], while the tensile strength of 250 × 10 × 2 mm^3^ samples was determined at a speed of 2 mm/min, following DIN EN ISO 527-4 [58].

### 3.2. Thermal Resistance

To induce defined damage states, 20 × 10 × 2 mm^3^ samples were exposed to a heat flux of 50 kW/m^2^ in several 5 s intervals, up to 35 s. This was achieved using an electric conical radiant heater within the Cone Calorimeter from Fire Testing Technology (East Grinstead, UK). Concurrently, the temperature on the back side of each sample was monitored with Ktype thermocouples (nickel-chromium/nickel) from Thermodirekt (Bruchköbel, Germany).

The thermal stability of the modified materials was evaluated by visualizing thermoinduced structural damage, such as delaminations down to the 10 μm range within the specimen volume, using the V-TOME XL 300 micro-computed tomography system from General Electric (Frankfurt am Main, Germany), equipped with a 180 kV microfocus X-ray source. To quantify these damages, a gray value analysis was applied. Additionally, matrix degradation was assessed by recording infrared spectra in the wavenumber range of 4000 to 400 cm^−1^ using the Tensor 27 from Bruker (Ettlingen, Germany). This system was outfitted with a Harrick ATR cell, a silicon crystal, and a pyroelectric detector.

## 4. Results and Discussion

### 4.1. Morphology of the Nanoparticles

Figure 2 depicts the magnified particle structures of unprocessed carbon black particles (CBPs) (a) and carbon nanotubes (CNTs) (b), along with their corresponding statistical size distributions. The CBPs (a) clearly exhibit a spherical morphology, with sizes ranging from 10 to 100 nm. Their particle size distribution follows a Gaussian curve, characterized by a Root Mean Square Error (RMSE) of 1.84. Consequently, 68% of the CBPs have a crosssectional size of 52 ± 25 nm. Due to their small particle size, they possess a high surface energy, leading to aggregation into larger clusters. In contrast, CNTs (b) extend to lengths in the micrometer range, with diameters limited to the nanometer scale. Their average cross-sectional size is only 25 ± 8 nm. This exceptional aspect ratio makes them prone to mechanical entanglement, while van der Waals forces facilitate mutual attraction. As a result, they form a randomly planar-aligned network structure. Thus, the two carbon-based particle types differ significantly in their nanostructure.

In the subsequent step, various mass concentrations of these nanoparticles are incorporated into the uncured RTM6 resin to assess their processability. Figure 3 illustrates the dynamic viscosity of the mixtures with integrated CBPs (a) and CNTs (b) as a function of temperature. In its unmodified intial state (0%), the viscosity is approximately 539 ± 2 Pa·s at room temperature. Under these conditions, significant internal friction among the molecules results in a highly viscous resin. As the temperature increases, the viscosity of the resin initially decreases due to the greater kinetic energy, which promotes molecular motion and facilitates their sliding movement. This enables the resin to infiltrate carbon fiber fabrics during the wet lay-up process. Practical experience has shown that efficient lamination is achieved at temperatures between 50 and 60 °C, resulting in viscosities below 10 Pa·s. At even higher temperatures, the viscosity may increase again, hindering infiltration and potentially leading to defects such as pores or misaligned fibers.

Simultaneously, the incorporation of nanoparticles into the resin results in an increased dynamic viscosity. For instance, at their respective maximum particle concentrations, viscosities of up to 1106 ± 5 Pa·s for CBPs (a) and even 1217 ± 7 Pa·s for CNTs (b) are achieved at room temperature. These elevated values can be attributed to the particles creating disruption points within the uncured matrix, leading to heightened internal friction. As expected, the viscosity decreases with rising temperature; however, this reduction is less pronounced compared to that of the unmodified matrix. Consequently, the fabrication of high-quality modified laminates is limited to mass concentrations of less than 15% CBPs (a) and 2% CNTs (b) in the matrix. In comparison, only low concentrations of CNTs can be utilized due to their nanoscale structure, which significantly impedes the sliding motion between the molecules.

In this context, nanocomposites with concentrations of 1, 3, 5 and 10% CBPs as well as 0.1, 0.5 and 1.5% CNTs in the polymer matrix were produced. Figure 4 presents SEM images of the non-modified (a) and modified material containing 1.5% CNTs (b) in the crosssectional direction. In both sections, the fiber plies (0°/90°) and the transition to the adjacent ply are clearly visible. Within the fabric ply (a1), the conductive carbon fibers are generally in very close contact with each other because, during the weaving process, the fiber strands are alternately positioned over and under the crossing strands. In contrast, the transitions between the plies (a2) are separated by an approximately 40 μm thick insulating matrix layer. However, by integrating conductive carbon nanotubes, this region can be bridged with conductive components. Interestingly, the nanotubes are relatively evenly dispersed between the fabric plies (b2), while they are agglomerated unevenly within the plies (b1). This incomplete infiltration arises from a combination of limited space between the fibers and the considerable length of the tubes. Nevertheless, additional junctions between the fibers and the tubes are formed within the fabric ply, further enhancing the conductive network. Overall, this creates a continuous network of conductive components that extends both laterally and vertically throughout the material. In the case of modification with carbon black, the nanoparticles cannot be visually distinguished from the polymer matrix, as they exhibit similar electron backscattering properties. Consequently, this SEM image does not visually differ from the unmodified material in Figure 4a. However, due to their spherical structure, a more homogeneous infiltration is anticipated, particularly within the fabric plies.

### 4.2. Characterization of the Nanocomposites

#### 4.2.1. Transport and Energy Properties

To gain a deeper understanding of the conductive network, Figure 5(a1) initially provides an assessment of electrical conductivity at the surface. The non-modified material displays a conductivity of only 10^−13^ S in the lateral direction, attributed to the electrical insulating nature of the matrix-enriched interfaces. Introducing low carbon black concentrations yields merely marginal improvements in electrical conductivity of CFRP. Only between 3 and 5% CBPs a significant increase to 10^−5^ S at the so-called percolation threshold is observed. Beyond this threshold, the system reaches a saturation, evidenced by an almost constant conductivity level. Similarly, the electrical conductivity of the bulk material (a2) displays comparable behavior in the vertical direction. The unmodified material exhibits a conductivity of 10^−11^ S/m over the cross-section. With increasing particle concentration, the conductivity also rises by a factor of 10^8^, indicating that the particle network acts equally in the lateral and vertical directions. In contrast, the addition of CNTs leads to a significant increase in conductivity at both the surface (b1) and in the bulk (b2), even at very low concentrations of 0.05 to 0.15%, due to their exceptionally high length-to-thickness ratio. Consequently, only 1.5% CNTs are required to achieve conductivity enhancements comparable to those observed in CBP-modified materials.

This progression of electrical conductivity can be schematically divided into three zones. In the insulation zone, up to approximately 3% CBPs (a) or 0.05% CNTs (b), the material remains electrically insulating, because the conductive nanoparticles are still widely separated in the matrix. Significant conductivity improvements are observed only in the percolation zone, where a continuous network of conductive nanoparticles and carbon fibers forms. This improvement arises from the partially semiconducting behavior and tunneling effect, which allows electrons to jump between conductive components in the matrix, even at minimal distances [59]. As a result, electron transport is considerably less hindered than in the non-modified material. In the conductive zone, the network continues to expand but has only a negligible effect on electrical conductivity.

The following analysis focuses on the thermal properties depicted in Figure 6. First, the influence of nanoparticles on thermal conductivity (a, d) along the cross-section of the composites is examined. The non-modified sample exhibits a conductivity of just 0.483 ± 0.002 W/(m·K) at room temperature, indicating that the material acts as a thermal insulator due to the polymer matrix. With an increasing concentration of carbon-based nanoparticles, thermal conductivity rises. Concretely, at the maximum concentration of CBPs (a), the conductivity increases by 18% to 0.569 ± 0.009 W/(m·K), while the incorporation of 1.5% CNTs (d) leads to an improvement of just 13%. The carbon-based nanoparticles in the material thus do not induce a percolation effect in the thermal conductivity. Unlike electrical conductivity, where electrons traverse a conductive network, heat transport in nanocomposites occurs via phonons—quantized lattice vibrations. Since the nanoparticles are connected only by van der Waals forces, a high thermal contact resistance arises at their interfaces, enhancing phonon scattering and thereby disrupting heat conduction in the network [60]. Consequently, the improvement in thermal properties remains modest compared to that in electrical conductivity.

The specific heat capacity (b, e) of the non-modified material is approximately 1.018 ± 0.014 J/(g·K) under standard conditions. The addition of up to 10% CBPs (b) leads to an increase in heat capacity of over 3%, while the capacities at low nanotube concentrations (e) remain relatively constant. This indicates that the sample with a high carbon black concentration (b) can store slightly more thermal energy before experiencing a temperature rise. In contrast, the integration of low CNT concentrations (e) contributes negligibly to thermal energy absorption. This suggests that a very high concentration of carbon-based nanoparticles is required for a significant increase in heat capacity.

Finally, the material density (c, f) is examined. The density of the unmodified composite material (c, f) is approximately 1.401 ± 0.001 g/cm^3^. The incorporation of up to 10% CBPs (c) and 1.5% CNTs (f) results in no significant change in density. Consequently, the addition of carbon-based nanoparticles enhances the thermal properties without increasing the weight of these specimens.

#### 4.2.2. Modeling of Thermal Properties

In the past, several models have been developed to predict the thermal conductivity of nanocomposites [60,61]. In the parallel model, it is assumed that each component contributes to the overall conductivity in proportion to its volume fraction, according to Equation (Equation 1): (1)$$\lambda_{\mathrm{mod}}=\phi_{f}\lambda_{f}+\phi_{m}\lambda_{m}+\phi_{p}\lambda_{p}$$
where $\lambda_{\mathrm{mod}}$, $\lambda_{f}$, $\lambda_{m}$ and $\lambda_{p}$, represent the thermal conductivity of the modified composite, the carbon fibers, the polymer matrix, and the nanoparticles, respectively, while ϕ denotes their volume fraction in the material. The parallel model generally exhibits the theoretical limit in these samples, as it assumes perfect contact between the particles in a fully percolating network [60].

In contrast, the series model assumes that there is no contact between the particles [60]. The contribution of the particles is therefore limited to the area surrounding them with matrix. Consequently, the conductivity of the modified materials can be described according to Equation (Equation 2): (2)$$\lambda_{\mathrm{mod}}=\frac{1}{\left(\phi_{f}/\lambda_{f}\right)+\left(\phi_{m}/\lambda_{m}\right)+\left(\phi_{p}/\lambda_{p}\right)}$$

It has been shown that most of the experimental results lie between the two models, whereby the serial model is closer to the experimental data than the parallel model [61]. To improve the prediction quality further, both the areas with contact and without contact between the particles must be taken into account. For this purpose, a model based on the rule of mixtures has been developed in this study, considering the effective conductivity of the nanoparticles within the polymer matrix, as described by Equation (Equation 3): (3)$$\lambda_{\mathrm{mod}}=\left(1-\omega_{p,l}\right)\lambda_{l}+\omega_{p,l}\cdot \lambda_{p,\mathrm{eff}}$$
where $\lambda_{l}$ refers to the experimental thermal conductivity of the non-modified laminate, $\lambda_{p,\mathrm{eff}}$ represents the calculated effective thermal conductivity of the nanoparticles within the composite, and $\omega_{p,l}$ denotes the mass fraction of the nanoparticles in the laminate. In contrast to the parallel and series models, this approach makes no theoretical assumptions but calibrates the real effective thermal conductivity. To this end, the thermal conductivity of the nanocomposites was measured at different temperatures and compared with the calculated effective values using Equation (Equation 3). Simultaneously, the effective values were adjusted according to the least squares method to achieve the closest alignment between experimental and calculated values, maximizing the coefficient of determination. Specifically, an effective thermal conductivity of 2.72 W/(m·K) was determined for CBPs with a coefficient of determination of $R^{2}$ = 0.885, while even 8.89 W/(m·K) was achieved for CNTs with $R^{2}$ = 0.854. These calculated effective conductivity values lie thus significantly below the theoretical ranges, which extend from 6 to 174 W/(m·K) for CBPs and from 2000 to 6000 W/(m·K) for CNTs [62]. These high theoretical ranges cannot be achieved in the nanocomposites because the particles do not form a sufficiently dense network, the carbon nanotubes are not aligned, and the particles contain defects and impurities. Most notably, considerable interfacial resistance between neighboring particles further limits thermal conductivity. Nevertheless, the effective conductivity values for CNTs are over three times higher than those of CBPs, indicating that CNTs fundamentally offer greater potential for enhancing heat conduction in nanocomposites than CBPs.

Similarly, the specific heat capacity and density can be calculated using the analogous model in Equation (Equation 3). For a comprehensive analysis, the calibration results are presented in the Appendix A in Figure A1, while a summary of the calibrated effective thermal properties is provided in Table 1. Both types of carbon-based particles exhibit comparable effective specific heat capacities of approximately 1.54 J/(g·K) and effective densities of 1.66 g/cm^3^. The calculated values are thus relatively high compared to typical theoretical values, as the nanoparticles are considered not as loose particles but surrounded by a polymer matrix. Consequently, the effective values approximate those of the RTM6 matrix.

To validate the calibrated effective values, Figure 6 also shows the modeled data with a 5% margin of error alongside the experimental data. This clearly demonstrates that the modeled values closely align with the experimental results. For example, by integrating 10% CBPs into the polymer matrix, which corresponds to 4.5% CBPs in the laminate, the thermal conductivity is calculated as $\lambda_{\mathrm{mod}}$ = (1 − 0.045) × 0.483 W/(m·K) + 0.045 × 2.72 W/(m·K) = 0.584 W/(m·K). This value is in good agreement with the experimental thermal conductivity of 0.569 ± 0.009 W/(m·K). Overall, the model accurately reflects the actual thermal conductivity (a, d), specific heat capacity (b, e), and density (c, f) of the nanocomposite.

Heat conduction is naturally described by the instationary, heat conduction equation in Equation (Equation 4) [63]. Thereby, the thermal diffusivity *a* determines the temperature distribution in the material, as it represents the ratio between the thermal conductivity λ and the heat storage $\rho \cdot c_{p}$.(4)$$\frac{\partial T(z,t)}{\partial t}=a\cdot \frac{\partial^{2}T\left(z,t\right)}{\partial z^{2}}\;\;\mathrm{with}\;\;a=\frac{\lambda }{\rho \cdot c_{p}}$$

To illustrate this, Figure 7 presents the thermal diffusivity for different concentrations of CBPs (a) and CNTs (b). The unmodified material exhibits a thermal diffusivity of 0.338 ± 0.004 mm^2^/s at room temperature. With increasing concentration of carbon-based particles, the thermal diffusivity rises in accordance to the model. For example, at a maximum concentration of 10% CBPs (a), a thermal diffusivity of 0.384 ± 0.008 mm^2^/s is achieved, while 1.5% CNTs (b) reach 0.380 ± 0.002 mm^2^/s. This indicates that the integration of these particles generally leads to a slightly faster temperature equalization in the material under thermal load, as heat is distributed more effectively throughout the sample. In direct comparison, both modifications show similar improvements; however, a closer examination reveals significant differences in thermal properties, as depicted in Figure 6. In the samples containing up to 10% CBPs, both thermal conductivity and specific heat capacity values increase, whereas in the materials with CNTs, only thermal conductivity improves due to the lower concentration of 1.5%. Consequently, although the thermal diffusivity is similar, samples with CBPs exhibit enhancements in both heat conduction and heat storage.

The heat distribution can be explained schematically, as shown in Figure 7. When a thermal load is applied, the heat flux *q* impinges on the sample surface. By incorporating conductive carbon-based nanoparticles, the matrix-enriched interfaces can be bridged; however, only a slight improvement in heat conduction is achieved compared to theoretical expectations. This limitation arises because the dispersates or agglomerates are surrounded by the matrix, resulting in their separation. Consequently, a high thermal resistance occurs, particularly at the interfaces between the particles, which is represented by the dotted lines. In contrast, conductive components in direct contact, such as particle agglomerates or carbon fibers, exhibit lower thermal resistance, illustrated by the solid line. Nonetheless, this total resistance remains significantly high compared to metallic alternatives, resulting in only moderate improvements in temperature distribution within fiber matrix composites.

#### 4.2.3. Mechanical Properties

Finally, the impact of carbon-based nanoparticles on mechanical performance of the composite materials is examined. For this, Figure 8 shows the loss factor from the dynamic mechanical analysis for samples with varying concentrations of CBPs (a) and CNTs (b) as a function of temperature. The progression of the loss factor can generally be divided into three distinct temperature ranges: Up to approximately 175 °C, the loss factor increases only slightly due to the material’s rigid, glassy state. It reaches its maximum value exclusively in the glass transition region, where high internal friction between the molecules occurs. From around 260 °C onwards, the loss factor begins to decrease as the sample undergoes thermal degradation, transitioning into a softened, more viscous state.

The glass transition temperature marks the transition of the polymer matrix from a hard to a soft state at the maximum loss factor. The non-modified material exhibits a glass transition temperature of approximately 222 ± 2 °C. In comparison, the modified materials with integrated CBPs (a) display a slightly lower glass transition temperature of 215 ± 1 °C, while those containing CNTs (b) remain almost unchanged. This reduced glass transition temperature suggests that a high filling level of CBPs may slightly impair the cross-linking density of the thermoset and thus the mechanical integrity [64].

To verify mechanical integrity, Figure 9 illustrates the normalized strengths for varying concentrations of CBPs (a) and CNTs (b) in the composite. The unmodified material shows an interlaminar shear strength of 60.0 ± 1.7 MPa, a compressive strength of 518 ± 13 MPa, and a tensile strength of 575 ± 19 MPa. The integration of up to 10% CBPs (a) or 1.5% CNTs (b) into the material does not lead to significant deterioration in mechanical properties. Both interlaminar shear and compressive strength are particularly sensitive to fiber-matrix bonding [31,33]. Since the strength of the modified materials remains constant under these mechanical loads, it indicates that the bond between the fiber and matrix is not adversely affected by the integrated particles. Conversely, tensile strength reflects how effectively the external forces can be absorbed by the fibers. A slight decrease is noted only in the sample with 10% CBPs (a), which can be attributed not to the high particle concentration, but rather to the manufacturing process. During wet laminating, minor deviations in fiber orientation may occur, significantly impacting the strength of these long tensile samples. Considering this, the mechanical properties remain largely unaffected by the modifications involving carbon-based nanoparticles. The incorporation of these particles into CFRP thus enhances thermal properties without compromising mechanical integrity. However, this also means that the reinforcing potential of the nanoparticles does not lead to an improvement in the mechanical properties of the composite. Building on these findings, the subsequent investigation will focus on evaluating how these improved properties of the nanocomposites endure under thermal loading.

### 4.3. Thermal Resistance of the Modified Materials

Figure 10 presents the temperature evolution *T*(*t*) on the back side of samples with different concentrations of CBPs (a1) and CNTs (b1) under a heat flux of 50 kW/m^2^. During one-sided thermal exposure, heat is absorbed on the front side and conducted to the cooler back side, forming a temperature gradient. In this scenario, the unmodified material reaches a back side temperature of 393 ± 2 °C within 35 s. In comparison, the samples with the highest particle concentration of CBPs (a1) only attain 362 ± 19 °C, while those with CNTs (b1) reach 385 ± 4 °C. For a more detailed analysis, Figure 10 also depicts the temperature differences between the original and modified materials with CBPs (a2) and CNTs (b2). It is evident that increasing carbon-based particle concentration results in a larger negative temperature difference, indicating slower heating processes. For example, samples containing 1% CBPs (a2) warm approximately 9 °C slower than the unmodified specimens over the entire loading period, whereas those with 10% CBPs (a2) exhibit already a 25 °C reduction in heating. In contrast, samples with nanotube concentrations of up to 0.5% (b2) show no significant temperature reductions; only at a concentration of 1.5% an average temperature decrease of 10 °C is observed. Thus, particularly in materials with high concentrations of CBPs, lower temperatures are archieved, suggesting an improved heat distribution.

In the following, the effect of heat exposure on thermo-induced damage in the samples is investigated. To this end, Figure 11 on the right displays a *yz*-computed tomogram, illustrating the original material after a thermal loading period of 30 s. This tomogram reveals a damage distribution along the cross-section, which can be categorized into three distinct regions: starting from the back side, no damage is observed in the third region, defined as $r_{\mathrm{III}}$. Structural damage, such as delaminations, first appears in the second region, $r_{\mathrm{II}}$, resulting from a combination of internal stresses and the decomposition of the polymer matrix [65]. Internal stresses arise from the differing thermal expansion properties of carbon fibers, which contract axially, and the polymer matrix, which expands in all directions. Matrix decomposition is generally induced when temperatures close to the glass transition temperature [34]. Once critical stress and a significant degree of matrix decomposition are reached, thermo-induced delaminations emerge. In the tomography scan, these damages can be distinguished from the intact polymer matrix, as they usually contain air or pyrolysis gases. Since these gaseous mixtures have a lower density than the solid material, they absorb less X-ray radiation, resulting in greater transparency. Consequently, damaged areas appear darker in the tomography image, while intact material is represented in lighter gray tones. In the first region, $r_{I}$, extensive damage is evident near the front side due to matrix depletion. This depletion arises from the thermo-induced decomposition of the polymer matrix into pyrolysis gases, which may either escape from the surface or remain trapped within the material.

To quantify the structural damage, a grayscale analysis can be performed on the *xy*tomograms, as exemplified on the left side of Figure 11. For this, cross-sectional images are generated, and the number of voxels along with their corresponding RGB color values are evaluated. In the RGB color space, grayscale values increase linearly from black (Red = 0, Green = 0, Blue = 0) to white (255, 255, 255). This linear scaling enables the three color channels to be combined into a single channel. The damaged area can be distinctly differentiated from the intact material by applying a color threshold for the damage state at *c* = 88. Beginning from the back side, almost the entire color distribution of the tomogram remains above this threshold. Only 1% falls below this threshold, which does not indicate structural damage but is rather reflects noise effects within the tomogram. Significant increases in lower color values are observed only in the second region, corresponding to the formation of 11 to 17% damage. In the G939 fabric structure, these damages extend along the fiber strands in two directions: (0°/90°) and (+45°/−45°). In the first region, lower color values continue to rise, with pronounced damage areas of 44% developing adjacent to the weaving points.

Figure 12a illustrates the quantified damaged area along the ply depth for various samples subjected to 30 s of thermal exposure. Under one-sided thermal loading, it is evident that damage significantly diminishes from the front to the back side of the unmodified material, as expected. However, pronounced damage peaks are observed between the individual plies, likely due to elevated internal stresses resulting from the alternating fiber orientations of (+45°/−45°) and (0°/90°). In contrast, the modified materials containing 1.5% CNTs, and particularly those with 10% CBPs, exhibit substantially lower levels of damage throughout the entire cross-section. To provide a comprehensive examination, Figure 12b displays the cumulative damaged areas along the ply depth as a measure of damaged volume relative to irradiation time. In the unmodified sample, structural damage does not manifest for the first 15 s. Following this period, the volume of damage escalates rapidly, ultimately reaching nearly one-third of the sample after 35 s. Conversely, the damaged volume in the sample with 1.5% CNTs is slightly less pronounced, while the sample with 10% CBPs experiences structural damage much later, after 25 s. Consequently, the samples with high concentrations of carbon black exhibit the least amount of damage.

To compare matrix degradation among the materials, infrared spectra were analyzed. Initially, the characteristic infrared bands of the various components were identified, as shown in Figure 13a. The spectra of the RTM6/G939 demonstrate the ν(-C=C) ring vibration in the aromatic structure of the epoxy resin at wavenumbers 1510 and 1610 cm^−1^ [66]. In the heated state, above the glass transition temperature, additional ν(-C=O) stretching vibrations associated with oxidation products, such as amides, ketones, or aldehydes, become evident in the range of 1630 to 1800 cm^−1^ [66]. In contrast, the modified materials containing CBPs and CNTs do not show significant differences in the intensities or regions of the infrared bands, allowing for a comparative analysis of their thermal behavior.

Figure 13b therefore presents the overall developments in band intensities for the epoxy resin (EP) and oxidation products (OX) on the surface of the non-modified material subjected to increasing thermal stress. During heating, the band intensity of the carbonyl groups (OX) increases due to thermo-oxidative reactions. At lower temperatures, specifically up to 80 °C, defects such as unreacted resin or functional groups from the curing process may oxidize [66]. As the temperature rises, oxidation progressively affects the aliphatic compounds within the epoxy resin [66]. Simultaneously, the intensity of the epoxy resin (EP) bands diminishes as a result of thermo-induced decomposition processes. In this degradation process, secondary alcohols may initially undergo dehydrogenation, followed by the cleavage of allylic amine chains, resulting in the formation of volatile components and char residues [67]. To quantify the matrix degradation, these opposing trends are exploited by calculating the ratio $I_{d}$ = $I_{{1800-1630\;\mathrm{cm}}^{-1}}$/$I_{{1510\;\mathrm{cm}}^{-1}}$.

In Figure 14, the quantified matrix degradation on the back side of samples with varying concentrations of CBPs (a) and CNTs (b) during thermal stress is illustrated. Initially, the infrared intensity at the material’s surface is approximately 0.99 ± 0.03. As thermal loading increases, the intensity of the unmodified material begins to decline after about 7 s, indicating the onset of matrix degradation. This is followed by a sharp decline in infrared intensity, which reaches only 0.11 ± 0.01 after 35 s. In contrast, the matrix degradation in the sample containing up to 1% CBPs (a) occurs slightly later. A significant delay of approximately 5.0 s is only observed at higher concentrations, where a denser conductive network of carbon fibers and carbon-based nanoparticles forms, allowing for more efficient heat dissipation within the material. Similarly, in the specimen with a concentration of 1.5% CNTs (b), resin decomposition is delayed by approximately 2.3 s. Consequently, the thermal stability regarding matrix degradation is approximately twice as high in the samples with CBPs compared to those with CNTs, due to the higher particle concentration.

Thermo-induced matrix degradation, combined with structural damage, can significantly weaken the fiber-matrix bonding in composite materials. This results in reduced mechanical load transfer between the fiber and matrix, potentially diminishing the material’s strength and stability [31,33]. The short beam shear test is a highly sensitive method for characterizing these effects via interlaminar shear strength, as, in addition to normal compressive and tensile stresses perpendicular to the laminate plane, shear stresses primarily develop between the plies. To illustrate this, Figure 15 presents the interlaminar shear strength (ILSS) of specimens with varying concentrations of CBPs (a1) and CNTs (b1) after one-sided thermal loading. The initial shear strength of the samples is approximately 63 ± 3 MPa. After around 16 s, the strength of the unmodified sample begins to decrease, whereas specimens containing carbon-based nanoparticles retain their mechanical integrity for a longer period. To quantify this improvement, the irradiation times of the modified samples were multiplied by various delay factors until their strength-time data were optimally aligned with the reference curve of the non-modified material, using the least-squares method. For a more detailed analysis, Figure A2 in the Appendix A provides the transformed reference curves. Figure 15 directly illustrates the calibrated delay factors that reflect the strength retention effect resulting from the incorporation of carbon-based nanoparticles into the CFRP. The results clearly show that the mechanical integrity is maintained up to 17% longer only beyond the identified percolation threshold of 3–5% for CBPs (a2). In contrast, the strength loss can only be delayed by a maximum of 7% above the threshold of 0.05–0.15% for CNTs (b2). As previously indicated by temperature and thermo-induced damage progression, the delay in mechanical strength loss is particularly pronounced in samples with high carbon black content.

## 5. Conclusions

This study introduces an approach to enhance the thermal resistance of fiber-reinforced composites while preserving their core lightweight attributes, such as low density and high strength. To this end, nanocomposites were fabricated by progressively incorporating higher concentrations of carbon black particles (CBPs) and carbon nanotubes (CNTs), thereby pushing the thermal resistance to its practical limits within the material’s inherent constraints. The modified materials were then assessed for their properties and thermal stability.

The comparison between these different carbon modifications reveals that the particle structure limits the maximum feasible concentration in the polymer matrix. Due to their spherical shape, CBPs can be incorporated up to 10% without compromising the composite’s key properties, while CNTs, with their high aspect ratio, are limited to 1.5%. Within the samples, these particles form agglomerates and dispersions, predominantly accumulating in resin-rich interlayers. From around 3% CBPs and 0.05% CNTs, a percolation threshold is achieved, establishing a conductive network of nanoparticles and carbon fibers. This structure boosts electrical conductivity by eight orders of magnitude, whereas thermal conductivity increases by only 18% at most. This modest thermal improvement arises from phonon-based heat transfer and significant thermal boundary resistance between agglomerated and matrix-embedded nanoparticles. Thus, contrary to theoretical expectations, the resulting thermal conductivity values remain comparatively low. To quantify these effects, a model was developed that accurately characterizes the impact of these nanoparticles using experimental data, resulting in effective thermal conductivities of about 2.7 W/(m·K) for CBPs and 8.9 W/(m·K) for CNTs. Although CNTs hold greater potential for improving thermal properties, their restricted concentration mainly benefits heat distribution, while higher concentrations of CBPs also increase heat capacity. As a result, modified samples heat more slowly than unmodified ones, delaying thermo-induced damage and enhancing mechanical integrity by up to 17% for CBPs and 7% for CNTs.

Compared to other methods, integrating carbon-based nanoparticles improves the thermal resistance of CFRP without compromising its essential properties. However, the complexity of the wet-layup production process raises the question of whether the effort is justified by the delay in mechanical integrity loss. The effectiveness of this modification ultimately depends on the specific application and loading scenario. For prolonged fire exposure, such as in an aircraft fire, the benefits are limited. Conversely, for short-term loads like a lightning strike, this delay can be crucial.

A potential avenue to further improve the thermal performance of CFRP lies in optimizing the interaction between carbon-based nanoparticles and the matrix. Enhancing interfacial adhesion, for instance, through surface functionalization, may increase the effectiveness of these nanoparticles. Additionally, incorporating different types of carbonbased fillers, such as CBPs and CNTs, might offer synergistic effects. The distinct sizes and shapes of these materials could facilitate the formation of a hierarchical percolation network, potentially increasing the number of connection points and enhancing heat transfer efficiency.

The developed model demonstrates high predictive accuracy by considering both areas with and without direct contact between the nanoparticles, thus accurately reflecting the actual conditions within the material. A key prerequisite for this accuracy is the precise experimental determination of the thermal properties of the fiber composite material, which serves as the foundation for the effective calibrated values. Consequently, the model is versatile and applicable to various nanocomposites and modification components.

The model indicates that CNTs have a significantly higher potential for enhancing heat distribution compared to CBPs. However, the increasing viscosity of the polymer melt limits the concentrations that can be effectively processed using the wet-layup method. A promising alternative is the hand layup of CNT sheets, known as buckypapers, in conjunction with prepreg plies to create nanocomposites that fully exploit the advantages of CNTs. This approach will be explored further in future research.

## Figures and Tables

**Figure 1 nanomaterials-15-00546-f001:**
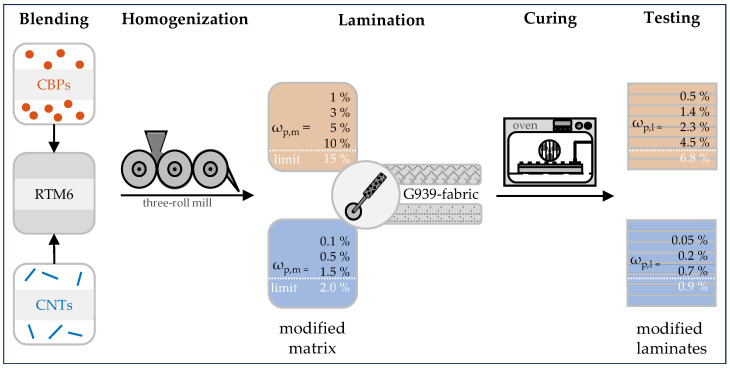
Schematic fabrication process of nanocomposites with integrated carbon black particles (CBPs) and carbon nanotubes (CNTs).

**Figure 2 nanomaterials-15-00546-f002:**
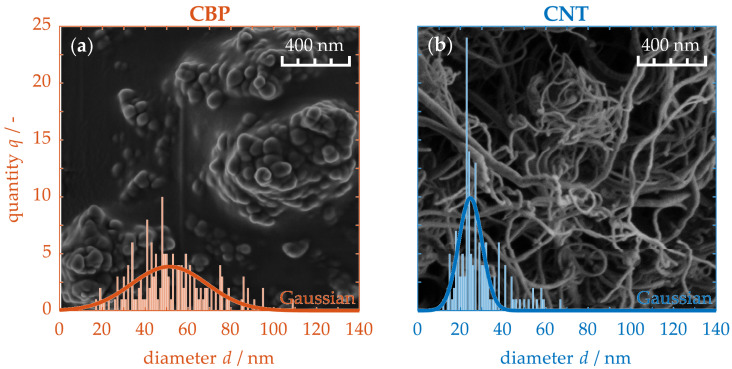
SEM micrographs of CBPs (**a**) and CNTs (**b**), including their statistical particle size distribution.

**Figure 3 nanomaterials-15-00546-f003:**
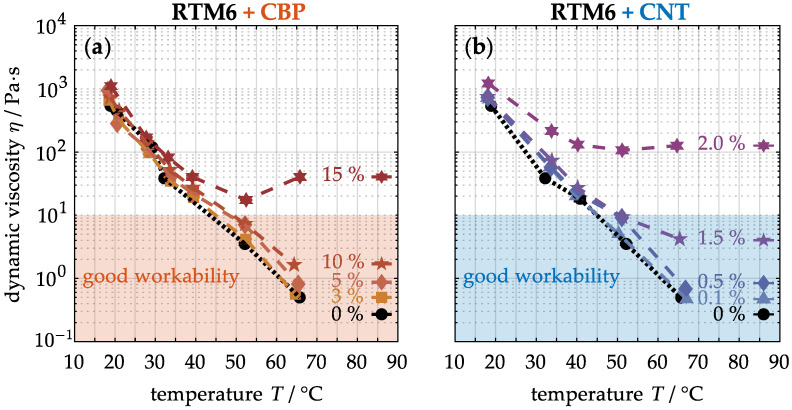
Dynamic viscosity of the uncured resin with different concentration of CBPs (**a**) and CNTs (**b**) with respect to temperature.

**Figure 4 nanomaterials-15-00546-f004:**
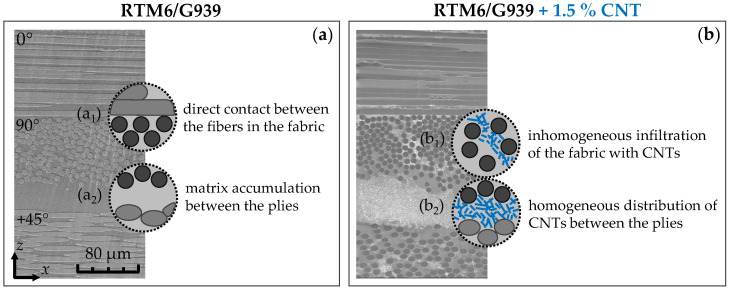
SEM micrographs along the cross-section of composites without nanoparticles (**a**) and with 1.5% CNTs (**b**).

**Figure 5 nanomaterials-15-00546-f005:**
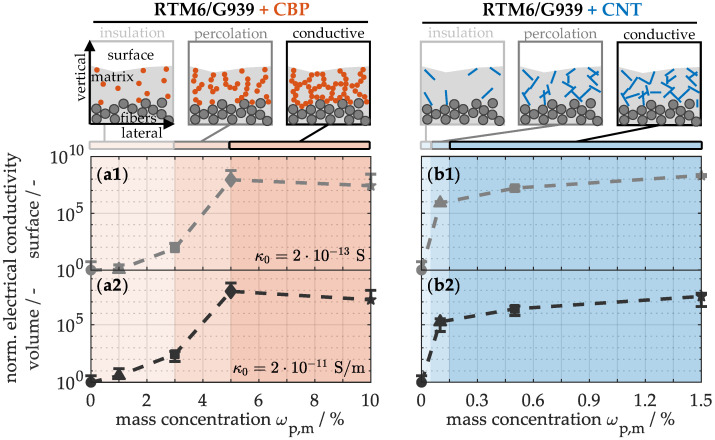
Normalized electrical conductivity on the surface and within the volume of nanocomposites with varying concentrations of CBPs (**a1**,**a2**) and CNTs (**b1**,**b2**), including a schematic representation of the percolation threshold.

**Figure 6 nanomaterials-15-00546-f006:**
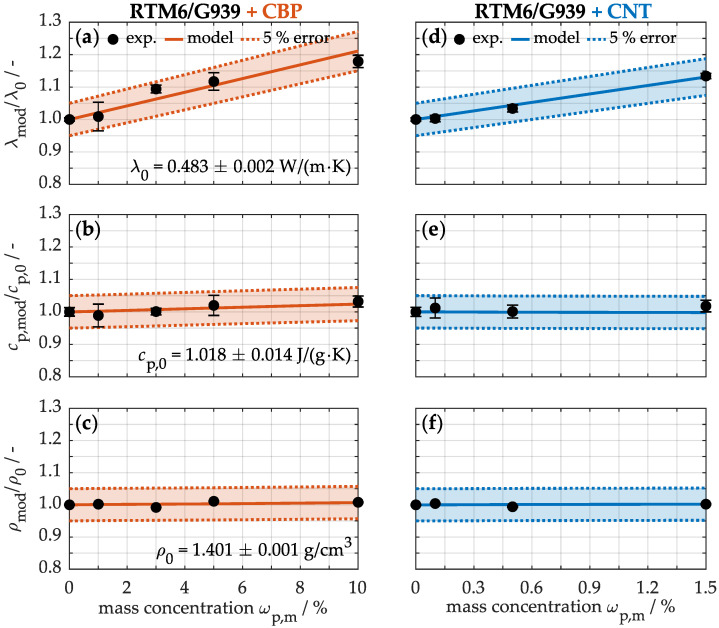
Experimental and calculated thermal conductivity λ (**a**,**d**), specific heat capacity $c_{p}$ (**b**,**e**) and density ρ (**c**,**f**) depending on the concentration of CBPs and CNTs. The calibration of the model is shown in Figure A1.

**Figure 7 nanomaterials-15-00546-f007:**
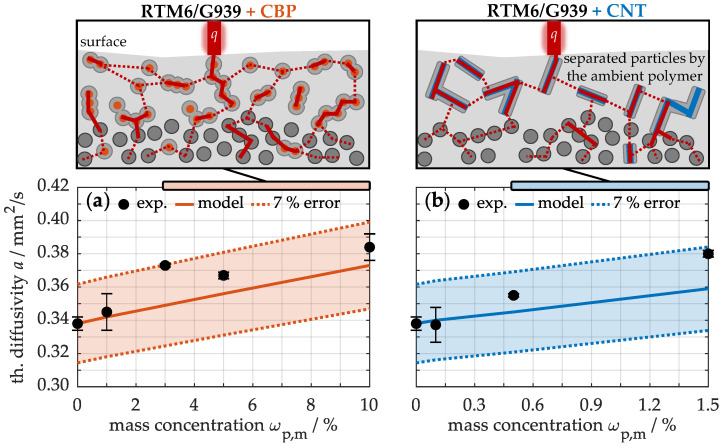
Experimental and calculated thermal diffusivity of laminates with different concentrations of CBPs (**a**) and CNTs (**b**), including a schematic illustration of heat conduction.

**Figure 8 nanomaterials-15-00546-f008:**
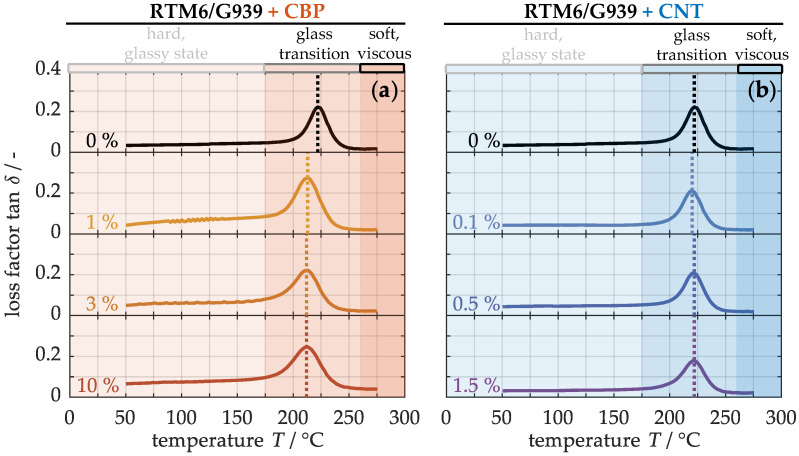
Loss factor of laminates with different concentrations of CBPs (**a**) and CNTs (**b**) in dynamic mechanical analyses.

**Figure 9 nanomaterials-15-00546-f009:**
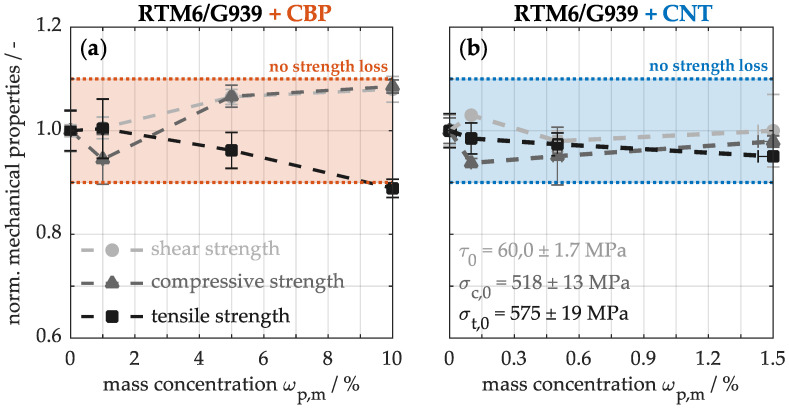
Mechanical properties of composite materials with varying concentrations of CBPs (**a**) and CNTs (**b**).

**Figure 10 nanomaterials-15-00546-f010:**
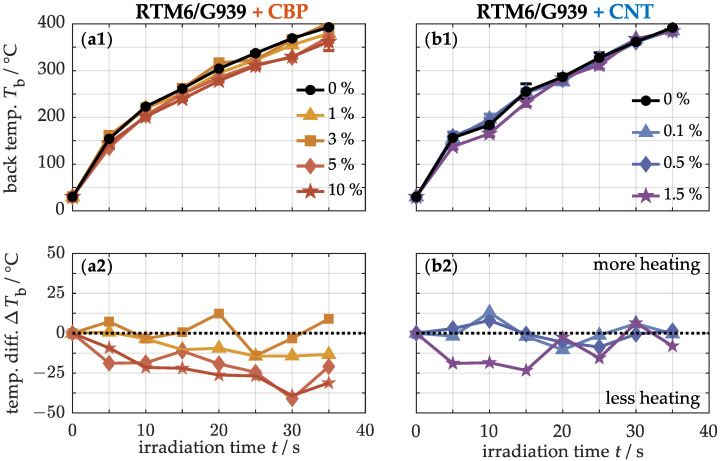
Back side temperature (**a1**,**b1**) and temperature difference between non-modified and modified materials (**a2**,**b2**) at different concentrations of CBPs and CNTs under one-sided thermal loads with 50 kW/m^2^.

**Figure 11 nanomaterials-15-00546-f011:**
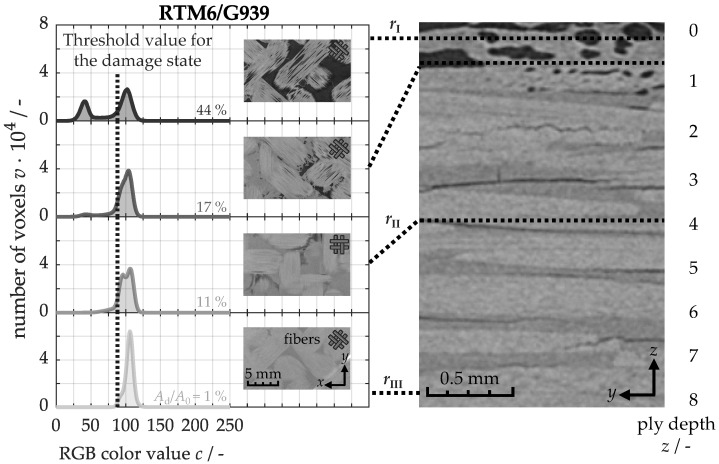
Gray value analysis of micro-computed tomograms of the non-modified material after a thermal loading period of 30 s at 50 kW/m^2^.

**Figure 12 nanomaterials-15-00546-f012:**
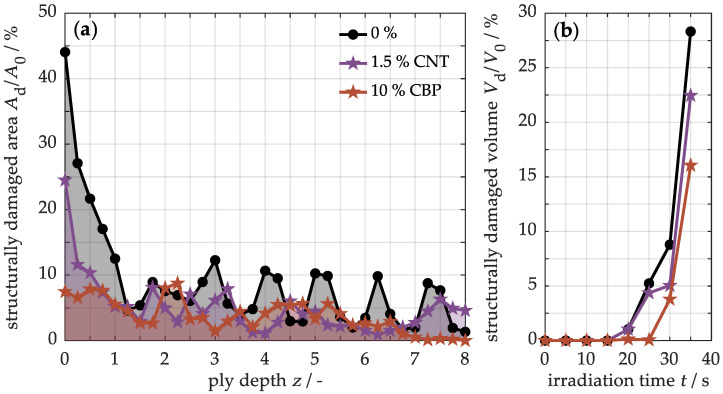
Quantification of the structurally damaged area along the cross-section after 30 s (**a**) and the damaged volume regarding the loading time (**b**) for samples with varying concentrations of CBPs and CNTs.

**Figure 13 nanomaterials-15-00546-f013:**
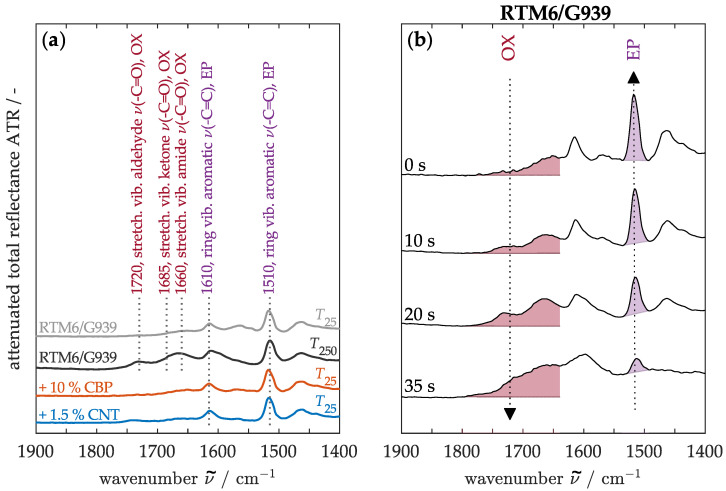
Infrared spectra of RTM6/G939 to visualize (**a**) the absorption bands and (**b**) the band intensity during thermal exposure at 50 kW/m^2^.

**Figure 14 nanomaterials-15-00546-f014:**
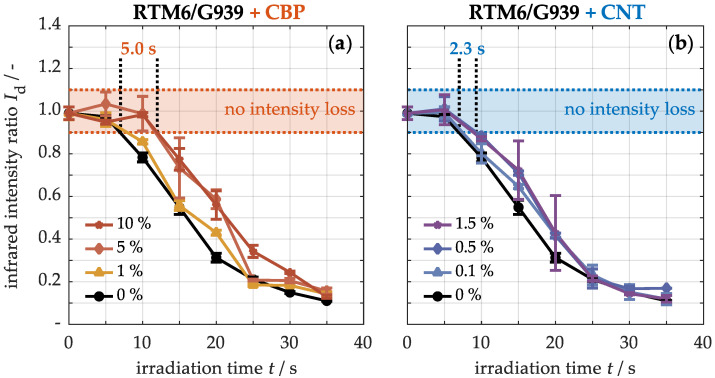
Quantification of the infrared band intensities of materials with different concentrations of CBPs (**a**) and CNTs (**b**) during thermal loading at 50 kW/m^2^.

**Figure 15 nanomaterials-15-00546-f015:**
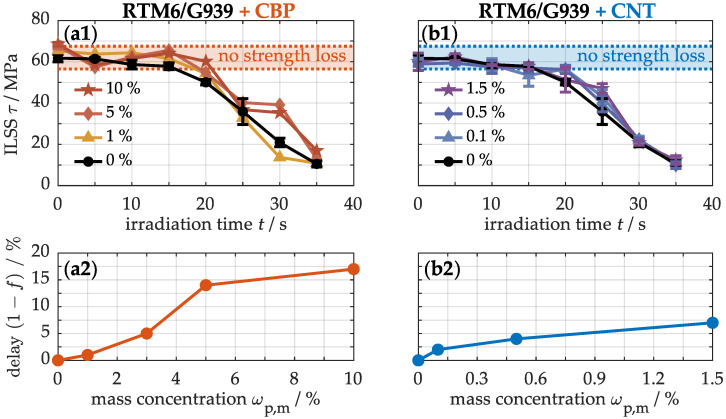
Interlaminar shear strength (**a1**,**b1**) and percentage delay in strength reduction between non-modified and modified materials (**a2**,**b2**) at different concentrations of CBPs and CNTs under one-sided thermal heating at 50 kW/m^2^. The transformation process is shown in Figure A2.

**Table 1 nanomaterials-15-00546-t001:** Calculated effective thermal properties of the carbon-based nanoparticles in CFRP. The calibration of the model is shown in Figure A1.

Effective Thermal Properties	CFRP	CBP	CNT
Thermal Conductivity/W/(m·K)	0.483 ± 0.002	2.72 ± 0.14	8.89 ± 0.44
Specific Heat Capacity/J/(g·K)	1.018 ± 0.014	1.55 ± 0.08	1.52 ± 0.08
Density/g/cm^3^	1.401 ± 0.001	1.61 ± 0.08	1.70 ± 0.09
Thermal Diffusivity/mm^2^/s	0.338 ± 0.004	1.09 ± 0.05	3.44 ± 0.17

## Data Availability

Data are contained within the article.

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
