# Peer review of "Pushing the Limits of Thermal Resistance in Nanocomposites: A Comparative Study of Carbon Black and Nanotube Modifications"

_nanomaterials, 2025, doi:10.3390/nano15070546_

Round 1
Reviewer 1 Report
Comments and Suggestions for Authors
This manuscript provides a method of enhancing the thermal conductivity of polymer composites by incorporating carbon black (CB) and carbon nanotubes (CNTs), thereby improving their thermal Resistance properties. This approach is considered promising for achieving enhanced thermal resistance in materials without significant weight addition or compromising mechanical performance. The study systematically investigates the mechanical, thermal, and rheological properties of RTM6/G939 resin composites modified with CB and CNTs, and employs mathematical modeling to simulate their behavior. The results demonstrate that adding up to 10% CB and 1.5% CNTs into the RTM6/G939 matrix can delay the thermal-induced mechanical property loss by 17% and 7%, respectively, which are notable and meaningful findings.
However, the following issues require clarification and revision in the manuscript:
1) Title Relevance: The current title "Pushing the Limits of Thermal Resistance in Nanocomposites" suggests that the results achieve "extremal" or "breakthrough" performance levels. However, the experimental data presented do not substantiate this claim. To align the title with the content, either:
- Provide additional evidence demonstrating that the achieved thermal resistance surpasses existing benchmarks (e.g., compare with state-of-the-art composites), or
- Explicitly state in the introduction/discussion that this work represents a "practical limit" given material constraints.
2) Dispersion Analysis: The surface characteristics ,including particle size, aspect ratio of CB/CNTs and their dispersion state in the polymer matrix significantly influence thermal/mechanical performance. However, the manuscript lacks enough characterization of filler surface properties and the description of the dispersion process (e.g., three-roll milling parameters:temperature, time).
3)Evidence of effective dispersion in the composite system (e.g., SEM images showing homogeneous distribution, absence of agglomeration).There are only raw filler SEM images provided, which do not confirm dispersion within the matrix.
Author Response
Thank you for your detailed and helpful review. I will now address your comments in detail and insert the corresponding revised sections. To provide clarity, I have also highlighted the changes in the text. I hope this meets your expectations.
Comments 1: Title Relevance: The current title "Pushing the Limits of Thermal Resistance in Nanocomposites" suggests that the results achieve "extremal" or "breakthrough" performance levels. However, the experimental data presented do not substantiate this claim. To align the title with the content, either:
- Provide additional evidence demonstrating that the achieved thermal resistance surpasses existing benchmarks (e.g., compare with state-of-the-art composites), or
- Explicitly state in the introduction/discussion that this work represents a "practical limit" given material constraints.
Response 1: I agree with you. To avoid any misunderstandings, I have explicitly mentioned in both the introduction and the conclusion that by integrating higher particle concentrations, the thermal resistance of the material is pushed to its practical limit.
Introduction: “By systematically increasing the particle concentration, the thermal resistance was pushed to the practical limits within the material's inherent constraints.”
Conclusion: “To this end, nanocomposites were fabricated by progressively incorporating higher concentrations of carbon black particles (CBPs) and carbon nanotubes (CNTs), thereby pushing the thermal resistance to its practical limits within the material's inherent constraints.”
Comments 2: Dispersion Analysis: The surface characteristics ,including particle size, aspect ratio of CB/CNTs and their dispersion state in the polymer matrix significantly influence thermal/mechanical performance. However, the manuscript lacks enough characterization of filler surface properties and the description of the dispersion process (e.g., three-roll milling parameters:temperature, time).
Response 2.1: I agree with you. The surface properties of the nanoparticles play a crucial role in the overall performance of the material. For this reason, I have incorporated all available information on this topic into the paper, although unfortunately, it is limited.
Materials: “For modification, two carbon-based nanomaterials were employed: First, PRINTEX® L6 carbon black particles (CBPs) [48] from Orion Engineered Carbons (Eschborn, Germany), characterized by a specific surface area of 21–1200 m²/g and no surface treatment [48]. Second, multi-walled carbon nanotubes (CNTs) of the type Baytubes® C70P [49] from Bayer MaterialScience AG (now Covestro AG, Leverkusen, Germany), with a carbon purity exceeding 95 % [50].”
The surface of the carbon black particles has not been functionalized. For this reason, I have suggested in the outlook the possibility that this could lead to improved thermal properties of the material performance.
Conclusion: “A potential avenue to further improve the thermal performance of CFRP lies in optimizing the interaction between carbon-based nanoparticles and the matrix. Enhancing interfacial adhesion, for instance, through surface functionalization, may increase the effectiveness of these nanoparticles.“
Furthermore, the bonding between the fiber and the modified matrix is crucial for the mechanical properties of the composite material. Since the strength remains unaffected even up to 10% CBPs and 1.5% CNTs, it can be assumed that the particles do not weaken this bonding.
Response 2.2: The description of the dispersion process has been updated with more precise details regarding temperature and time.
Materials: “The particle agglomerates in these mixtures were fractured using the EXAKT 80 E three-roll mill from EXAKT Advanced Technologies GmbH (Norderstedt, Germany). The process involved two cycles with a roller gap of 5 $\mu$m, varying roller velocities, and no roller heating, over a duration of approximately 20 minutes \cite{RN35}. The resulting suspensions were then degassed using the DAC 600.2 VAC-P speed mixer at a maximum frequency of 2350 rpm and a vacuum of 10 mbar at room temperature for three minutes.”
Comments 3: Evidence of effective dispersion in the composite system (e.g., SEM images showing homogeneous distribution, absence of agglomeration).There are only raw filler SEM images provided, which do not confirm dispersion within the matrix.
Response 3: In Figure 4, I show SEM images of the cross-section of (a) CFRP (+CB) and (b) CFRP + CNT. In (b), the mostly homogeneous distribution of CNTs between the fabric plies can be observed, while within the plies, the CNTs tend to agglomerate unevenly due to their high aspect ratio. In (a), we unfortunately could not visualize the distribution of carbon black in the composite, as these particles exhibit similar backscattering properties to the polymer matrix. However, it is assumed that a more uniform distribution is achieved due to the round shape of the particles.

Reviewer 2 Report
Comments and Suggestions for Authors
comment with minor revision.
The author had studiedCBPs and CNTs separately, but hybrid filler systems (CB + CNT) often exhibit synergistic thermal conductivity improvements due to hierarchical percolation networks. Did the authors explore potential synergistic effects between the two?
2-The thermal performance of CBPs with. CNTs are strongly dependent on interfacial adhesion. Was functionalization considered to improve nanoparticle-matrix bonding?
Author Response
Thank you for your helpful review. I will now address your comments in detail and insert the corresponding revised sections. To provide clarity, I have also highlighted the changes in the text. I hope this meets your expectations.
Comments 1: The author had studiedCBPs and CNTs separately, but hybrid filler systems (CB + CNT) often exhibit synergistic thermal conductivity improvements due to hierarchical percolation networks. Did the authors explore potential synergistic effects between the two?
Response 1: In this study, the initial focus was on investigating the effects of CB and CNT individually within the system. I agree with you that the synergistic effect between the two particles could potentially lead to significant improvements in thermal properties. The combination of both particles could be further investigated in a future study. For this reason, I have highlighted this possibility as part of the outlook.
Conclusion: „Additionally, incorporating different types of carbon-based fillers, such as CBPs and CNTs, might offer synergistic effects. The distinct sizes and shapes of these materials could facilitate the formation of a hierarchical percolation network, potentially increasing the number of connection points and enhancing heat transfer efficiency.“
Comments 2: The thermal performance of CBPs with. CNTs are strongly dependent on interfacial adhesion. Was functionalization considered to improve nanoparticle-matrix bonding?
Response 2: I agree with you that the thermal performance is determined by the surface properties of the particles. In this study, the CBPs and CNTs were used directly from the manufacturer without further treatment. The available surface properties of the particles have been included. According to the manufacturer, the CBPs were not functionalized, which is why I have included the possibility of surface functionalization in the outlook.
Materials: “For modification, two carbon-based nanomaterials were employed: First, PRINTEX L6 carbon black particles (CBPs) \cite{RN48} from Orion Engineered Carbons (Eschborn, Germany), characterized by a specific surface area of 21–1200 m²/g and no surface treatment \cite{RN48}. Second, multi-walled carbon nanotubes (CNTs) of the type Baytubes C70P \cite{RN49} from Bayer MaterialScience AG (now Covestro AG, Leverkusen, Germany), with a carbon purity exceeding 95 \% \cite{RN71}.“
Conclusion: „A potential avenue to further improve the thermal performance of CFRP lies in optimizing the interaction between carbon-based nanoparticles and the matrix. Enhancing interfacial adhesion, for instance, through surface functionalization, may increase the effectiveness of these nanoparticles.“

Round 2
Reviewer 1 Report
Comments and Suggestions for Authors
The revision is acceptable.